# Cohort profile: Scotland's record-linkage e-cohorts of people with intellectual disabilities, and autistic people (SCIDA)

Sally-Ann Cooper , Angela Henderson , Deborah Kinnear, Daniel Mackay, Michael Fleming, Gillian S Smith , Laura Anne Hughes-McCormack , Ewelina Rydzewska, Kirsty Dunn, J P Pell, Craig Melville

Institute of Health and Wellbeing, University of Glasgow, Glasgow, UK

**Correspondence to**
Professor Sally-Ann Cooper;
Sally-Ann.Cooper@glasgow.ac.uk

## ABSTRACT

**Purpose** To investigate health, mortality and healthcare inequalities experienced by people with intellectual disabilities, and autistic people, and their determinants; an important step towards identifying and implementing solutions to reduce inequalities. This paper describes the cohorts, record-linkages and variables that will be used.

**Participants** Scotland's Census, 2011 was used to identify Scotland's citizens with intellectual disabilities, and autistic citizens, and representative general population samples with neither. Using Scotland's community health index, the Census data (demography, household, employment, long-term conditions) were linked with routinely collected health, death and healthcare data: Scotland's register of deaths, Scottish morbidity data 06 (SMR06: cancer incidence, mortality, treatments), Prescribing Information System (identifying asthma/chronic obstructive pulmonary disease; angina/congestive heart failure/hypertension; peptic ulcer/reflux; constipation; diabetes; thyroid disorder; depression; bipolar disorders; anxiety/sleep; psychosis; attention deficit hyperactivity disorder; epilepsy; glaucoma), SMR01 (general/acute hospital admissions and causes, ambulatory care sensitive admissions), SMR04 (mental health admissions and causes), Scottish Care Information–Diabetes Collaboration (diabetic care quality, diabetic outcomes), national bowel screening programme and cervical screening.

**Findings to date** Of the whole population, 0.5% had intellectual disabilities, and 0.6% were autistic. Linkage was successful for >92%. The resultant e-cohorts include: (1) 22 538 people with intellectual disabilities (12 837 men and 9701 women), 4509 of whom are children <16 years, (2) 27 741 autistic people (21 390 men and 6351 women), 15 387 of whom are children <16 years and (3) representative general population samples with neither condition. Very good general health was reported for only 3389 (15.0%) people with intellectual disabilities, 10 510 (38.0%) autistic people, compared with 52.4% general population. Mental health conditions were reported for 4755 (21.1%) people with intellectual disabilities, 3998 (14.4%) autistic people, compared with 4.2% general population.

**Future plans** Analyses will determine the extent of premature mortality, causes of death, and avoidable deaths, profile of health conditions and cancers, healthcare quality and screening and determinants of mortality and healthcare.

## STRENGTHS AND LIMITATIONS OF THIS STUDY

⇒ Ninety-four per cent Census completion for a whole country population, and >92% successful record linkage.
⇒ Record linkage across multiple data sources.
⇒ Limitations include the use of routinely collected data.
⇒ Reliance on recorded cause of death on death certificates.
⇒ Self/proxy reporting of intellectual disabilities and autism, and other Census data.

## INTRODUCTION

People with intellectual disabilities are thought to die about 20 years earlier than other people,[1] or 28 years earlier for people with Down syndrome,[2] but most studies are small. A large English study found a standardised mortality ratio (SMR) of 3.18 (95% CI 2.94 to 3.43).[3] Two large Scottish studies reported an SMR of 2.24 (95% CI 1.98 to 2.49) in adults overall; 5.28 (95% CI 3.98 to 6.57) for Down syndrome adults, and 1.93 (95% CI 1.68 to 2.18) for adults without Down syndrome,[4] and an SMR of 11.6 (95% CI 9.6 to 14.0) in children overall.[5] Autism is also thought to be associated with premature death, though there have been few studies, and co-occurring intellectual disabilities may account for some of the reported findings.[6–9] The most common causes of death of people with intellectual disabilities are likely to differ from other people. For example, aspiration pneumonia which is avoidable, may occur quite commonly,[3 4 10] and SMRs have been reported as 2.4 (95% CI 1.3 to 3.8) for colorectal cancer, and 2.3 (95% CI 1.0 to 4.3) for female genital cancers.[3] Additionally, probably about 40% of early mortality would have been treatable by good quality healthcare, which may be double the amount that occurs in other people.[4 10 11] Causes of death of autistic people may also differ from those

in the general population, with deaths related to the nervous system, and injury and poisoning reported to be the most common causes.[8] These inequalities are a major injustice that need addressing. An important step towards this is to gain a better understanding of health, mortality, healthcare and their determinants in people with intellectual disabilities and autistic people.

The health profiles of people with intellectual disabilities and autistic people differ from other people, which has a bearing on mortality. People with intellectual disabilities have, for example, high rates of epilepsy, mental illness, gastrointestinal disease, antipsychotic use and polypharmacy.[12–14] They may have higher diabetes rates,[15 16] due to sedentary lifestyles, obesity and antipsychotics.[17 18] Cancer profiles appear to differ from other people, with colorectal, stomach, oesophageal, brain and uterine cancer, and leukaemia all having been reported to have higher incidence in people with intellectual disabilities, but with some inconsistencies between studies, and wide CIs due to study sizes.[19–21] Higher incidence of gastrointestinal cancers is plausible, due to poor diets, lifestyle and high rates of *Helicobacter pylori* and gastro-oesophageal reflux disorder, and lower screening uptake. Multimorbidity is typical in people with intellectual disabilities,[13 14] and poor health is reported for people with intellectual disabilities across the full life-course.[22] In autistic people, general health is also reported to be poor across the lifespan.[23 24] Autistic people commonly have comorbid physical conditions, and some are more prevalent than in the general population, including sleep problems, epilepsy, sensory impairments, atopy, autoimmune disorders and obesity, but not asthma.[25] However, there are substantial gaps in the evidence-base on the health of autistic people, and some findings are inconsistent.[25]

Despite this higher burden of health need and early mortality, healthcare has been reported to be poorer for people with intellectual disabilities,[26] and for autistic people,[27] and barriers in access to healthcare have been reported.[26 27] Adults with intellectual disabilities are reported to participate less in cervical screening and breast screening,[28] and bowel screening[29] than other people. There is also some indication that people with intellectual disabilities and autistic people are more likely to be hospitalised for ambulatory care-sensitive conditions than other people; these are conditions that should be fully manageable in primary healthcare settings, and so such hospitalisations are a further marker for poor care.[30 31] These factors may impact on poorer outcomes and survival, due to later stage disease when treatment is initiated, and poorer management of disease.

Internationally, one of the challenges in investigating these issues at a population level is the lack of data on representative samples of people with intellectual disabilities, or autistic people. Important studies have used novel methods to link routinely collected administrative data and identify people with intellectual disabilities via their use of educational services, services designed for adults with intellectual disabilities and/or though identifying

diagnostic codes in health records likely to indicate the presence of intellectual disabilities or autism.[32–35] These approaches provide valuable data and also challenges, as, after leaving education/school, not all adults with intellectual disabilities use intellectual disabilities services, and nor do autistic adults, and there are coding issues in health records which are completed by multiple professionals and may be both incomplete and overinclusive for syndromes that are associated with intellectual disabilities in some but not all people. Additionally, caseness may be variably defined, with the term intellectual and developmental disorders being commonly used and which may or may not be a broader definition of neurodevelopmental disorders combining, for example, some or all of intellectual disabilities, autism, attention deficit hyperactivity disorder and specific learning disabilities.[35] A further approach to population studies is using national surveys conducted with samples of the population, but has the limitation that people with intellectual disabilities or autism are represented in only small numbers and so restricts the analyses that can be undertaken, and in longitudinal research they tend to have higher cohort attrition and more biased cohort retention than other people.[35 36] In Scotland, the national Census in 2011 included questions on both intellectual disabilities and on autism, and so presents an ideal data-source for research with these populations. We are not aware of any other national Census data which has included data on intellectual disabilities and autism, other than Irish data which has the limitation of not distinguishing intellectual disabilities from specific learning disabilities. Record-linkage of Scotland's Census, 2011 to Scotland's other rich routinely collected health data sets and national statistics provides an excellent vehicle for detailed investigation of these issues. Scotland's Census, 2011 systematically questioned whether every person in the country had intellectual disabilities, and/or autism, and distinguished intellectual disabilities from dyslexia and other specific learning disabilities. It included data on 94% of the whole Scottish population, and imputed the remaining 6% (5 295 403 people in total). Due to its large size, it has power, when linked to other databases to answer important questions on health, causes of mortality, determinants and healthcare in people with intellectual disabilities, and autistic people. This is an important step leading towards a reduction in morbidity, early mortality and avoidable deaths and improvements in healthcare of people with intellectual disabilities and autistic people.

In Scotland's Census, 2011, of the whole Scottish population, 0.5% were recorded as having intellectual disabilities (0.4% women and 0.6% men), and 0.6% were recorded as being autistic people (0.3% women and 1.0% men). Regarding autism, 1.6% of all Scottish children and young people (<25 years of age) were recorded to have autism (0.7% women and 2.5% men), while 0.2% of adults aged 25 years and over were recorded to have autism (0.1% women and 0.3% men), reflecting the diagnostic broadening of autism spectrum disorders in

recent years, and greater awareness of the condition. This also accounts for why, of the autistic children and young people, 15.0% additionally had intellectual disabilities, while of the autistic adults aged 25 years and over, 29.4% additionally had intellectual disabilities. In total, 1.1/1000 people had both autism and intellectual disabilities, and are represented in both the intellectual disabilities and the autism cohorts. There is some evidence to suggest that autism is less recognised in women than in men,[37] but less so for people with more severe intellectual disabilities.[38]

We created two data sets, with record linkage to several of Scotland's health data sets. Linkage was undertaken for the 94% of the Scottish population who completed the Scottish Census, 2011. Of the people with intellectual disabilities recorded in Scotland's Census, 2011, 22 538 (92.9%) were successfully linked with their health records. Of the people with autism recorded in Scotland's Census, 2011, 27 741 (94.6%) were successfully linked with their health records. Regarding the general population who had neither intellectual disabilities nor autism, of the 15% randomly selected, 700 437 (95.1%) were successfully linked to their health records; while of the 5% randomly selected, 233 378 (95.1%) were successfully linked to their health records. Hence, each data set includes data on 22 538 people with intellectual disabilities, 27 741 autistic people, and a randomly selected comparison general population with neither intellectual disabilities nor autism. For ethical reasons, two data sets were necessary to ensure data minimisation, as our research questions on cancer incidence and mortality required a larger comparison group to reach adequate power, due to lower incidence, than was required to answer the other research questions. The minimal number of variables was requested for each data set: in data set 1, this therefore did not include ethnicity, nor general or mental health variables, as there is not sufficient power to investigate these factors in relation to causes of deaths and cancers.

► Data set 1 includes 22 538 people with intellectual disabilities, 27 741 autistic people and a random sample of 700 437 (15%) of the Scottish population. It will be used to answer research questions on SMRs, causes of deaths, treatable deaths, cancer standardised incidence ratios (SIRs) and cancer SMRs.

► Data set 2 includes 22 538 people with intellectual disabilities, 27 741 autistic people and a random sample of 233 378 (5%) of the Scottish population. It will be used to answer research questions on independent predictors of deaths and treatable deaths, comparison of uptake of cervical screening and bowel screening, descriptions of population health burden and quality of healthcare.

The sample size for the comparison sample in data set 1 was determined using the Scottish crude annual incidence rates and mortality rates for bowel cancer, oesophageal cancer and uterine cancer as examples, and previously published SIRs and SMRs for people with intellectual disabilities.[3 19 21] The figure 1 shows examples

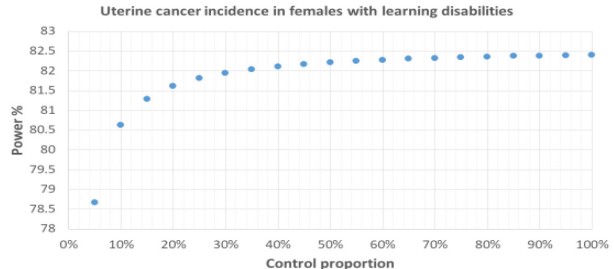

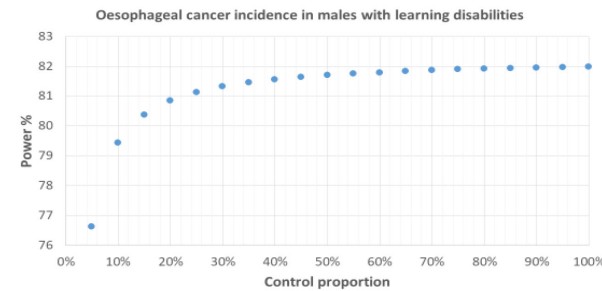

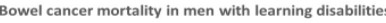

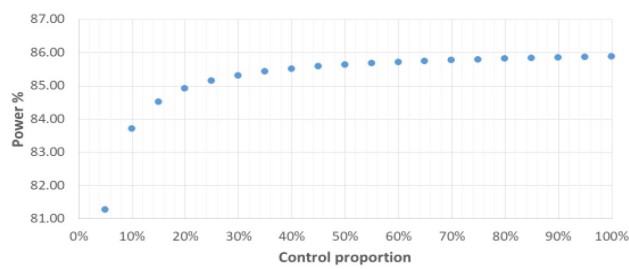

**Figure 1** Example power calculations across comparison sample sizes.

of some of the power graphs produced which led to the decision to use 15% of the Scottish population as the comparison group.

The data sets we linked together and variables we selected were based on these specific research questions posed on (a) people with intellectual disabilities, and (b) autistic people, compared with other people:

1. What are the age and sex-SMRs?
2. What are the common causes of death, and what proportion are treatable deaths?
3. Do cancer incidence and mortality differ?
4. What personal (including diseases identified via medication and admissions), household and service-related

factors (including hospital admissions) are markers/predictors of deaths and treatable deaths?

5. Do uptake of the national bowel and cervical cancer screening programmes differ?
6. What is the population health burden (identified via medication and admissions)?
7. Do people with diabetes* or other ambulatory care-sensitive conditions, receive as good healthcare, and are diabetes or other ambulatory care-sensitive conditions more likely to result in-hospital admissions and deaths?

*Diabetes was selected as an exemplar due to Scotland's Register of Diabetes Care.

The purpose of this report is to describe the e-cohorts we have created, the record-linkages that have been made, and the variables that are available within the data sets to investigate the above research questions.

## COHORT DESCRIPTION

Data sharing agreements are in place with the data controllers of all the linked data sets.

Table 1 shows the data sources which were linked, the dates of the data collection, the intended research purposes for the data and the data controllers.

**Table 1** Linked data sources included in the two data sets

| Resource | Time period | Use | Data controller |
|---|---|---|---|
| **Data set 1** | | | |
| Scotland's Census, 2011 | 27.3.11 | To identify the populations with intellectual disabilities, and autism, and the comparison population, and their basic demography. | National Records of Scotland. |
| Register of Deaths | 27.3.11–15.08.20 | To identify deaths, causes of deaths and treatable deaths. | National Records of Scotland. |
| Scottish Morbidity Records 06 | 27.3.11–15.08.20 | To identify cancer incidence, mortality and treatments (quality of healthcare). | NHS National Services Scotland, Information Services Division. |
| **Data set 2** | | | |
| Scotland's Census, 2011 | 27.3.11 | To identify the populations with intellectual disabilities, and autism, and the comparison population, their basic demography, household, employment and long-term conditions data (potential determinants of mortality). | National Records of Scotland. |
| Register of Deaths | 27.3.11–15.08.20 | To identify deaths. | National Records of Scotland. |
| Prescribing Information System | 27.3.11–27.9.11 | To identify people with: asthma/chronic obstructive pulmonary disease; angina/congestive heart failure/hypertension; peptic ulcer/reflux; constipation; diabetes; thyroid disorder; depression; bipolar disorders; anxiety/sleep; psychosis; ADHD; epilepsy; glaucoma (health burden). | NHS National Services Scotland, Information Services Division. |
| Scottish Morbidity Records 01 | 27.3.11–15.08.20 | To identify general/acute hospital admissions and causes, and ambulatory care sensitive admissions. | NHS National Services Scotland, Information Services Division. |
| Scottish Morbidity Records 04 | 27.3.11–15.08.20 | To identify mental health admissions and their causes. | NHS National Services Scotland, Information Services Division. |
| SCI-diabetes | 27.3.11–15.08.20 | To identify quality of diabetic care and diabetic outcomes (quality of healthcare). | Caldicott guardians. |
| National Bowel Screening Programme | 27.3.11–15.08.20 | To identify screening uptake. | NHS National Services Scotland, Information Services Division. |
| Cervical screening | 2016–15.08.20 | To identify screening uptake. | NHS National Services Scotland, Information Services Division. |

ADHD, attention deficit hyperactivity disorder; NHS, National Health Service; SCI-diabetes, Scottish Care Information - Diabetes Collaboration.

The variables we selected to include from each of the data sources were outcomes, exposures or potential confounders. A table of the full list of variables is included in online supplemental file 1. Data linkage of Scotland's Census, 2011, and de-identification, were carried out by the National Records of Scotland (NRS) Indexing Service (a Trusted Third Party Linkage Service) to the NRS Indexing Spine, which includes each person's Community Health Index. The Community Health Index is the unique National Health Service identifier given to everyone in Scotland; hence this indexing then enabled linkage to all the other selected health databases. Online supplemental file 2 shows the process. The linked data sets are held in Scotland's national safe haven, where access is provided to the approved members of our team.

Scotland's Census, 2011 includes information on people staying in Scotland on 27 March 2011. It includes people in both private households and communal establishments (eg, care homes). One householder on behalf of all occupants in private households, or the manager on behalf of all occupants of communal dwellings, was required to complete the Census questionnaire. Non-responders were followed-up, and help was provided to complete the form if needed. It was a legal requirement to complete the Census questionnaire, and a statement on the form said that non-completion, or supplying false information could attract a £1000 fine. It was estimated to have achieved a 94% response rate.[39] During the original data processing, the Census team adjusted for the 6% for whom data were missing, using a Census Coverage Survey (including around 40 000 households) to estimate numbers and characteristics. Self/proxy reporting was used to identify people with intellectual disabilities, autistic people and people with other long-term conditions from the Census question 20: 'Do you have any of the following conditions which have lasted, or are expected to last, at least 12 months? Tick all that apply'. Respondents were given a choice of 10 response options: (1) deafness or partial hearing loss, (2) blindness or partial sight loss, (3) learning disability (eg, Down's syndrome), (4) learning difficulty (eg, dyslexia), (5) developmental disorder (eg, autistic spectrum disorder or Asperger's syndrome), (6) physical disability, (7) mental health condition, (8) long-term illness, disease or condition (9) other condition, (10) no condition. Cognitive question testing with retrospective probing was undertaken on question 20 prior to the Census, to test whether the questions were answered accurately and willingly by respondents, and to identify what changes might be required to improve data quality and/or the acceptability of the response options. The pilot testing was conducted with 102 participants with a mix of gender and age, both with and without health conditions/disabilities (including people with more than one of the conditions), and included people with autism, intellectual disabilities, dyslexia, dyspraxia, speech impairment, mental health conditions (both milder and more serious) and other long-term conditions. This resulted in a redesign of the question on autism, to the version included in the actual Census, to accurately capture the data on autism, in keeping with feedback from the pilot testing with people with autism. (Henceforth, throughout this paper we refer to 'autism', rather than the exact wording on the Census described above). Importantly, question 20 distinguished autism from learning disability (which in the UK is synonymous to the international term 'intellectual disabilities'), learning difficulty (which in the UK is synonymous to the international term 'specific learning disability' such as dyslexia) and mental health conditions. All people with a positive response to question 20 on intellectual disabilities and/or autism were included in our record-linked data sets.

The data sources linked to the variables from Scotland's Census, 2011 are described below. All provide routinely collected data for the whole of Scotland. The Scottish Morbidity Record (SMR) routinely records information from episodes of healthcare.

1. SMR 01 provides data on general/acute hospital admissions and causes using the international statistical classification of diseases and related health problems, 10th edition (ICD-10) codes.[40]
2. SMR 04 provides data on mental health admissions and causes also using ICD-10 codes.
3. SMR 06 is the Cancer Registry and provides information on the disease, its treatments, and outcomes.
4. Prescribing Information System provides information on all medicines that are prescribed and dispensed in the community in Scotland, or issued in hospitals and dispensed in the community, and prescriptions written in Scotland that were dispensed elsewhere in the UK, using British National Formulary codes.
5. Scottish Care Information–Diabetes Collaboration is a national diabetes register that collects data on the quality of care, and outcomes for all patients with diabetes.
6. The national bowel, and cervical screening programmes collect data on screening uptake.
7. Scotland's national death register collates information from death certificates including cause of death as recorded by the certifying clinician, using ICD-10 codes.

Table 2 reports the characteristics of the cohorts with intellectual disabilities, and autism, and the random samples of the general population with neither of these conditions.

### Patient and public involvement

This resource and the programme of research it will support, was designed to respond to the growing concern expressed by people with intellectual disabilities, autistic people, their families, and third sector organisations about premature deaths, and poor healthcare. The Scottish Learning Disabilities Observatory, where this research is being undertaken, has a specific remit for such work. Its steering group includes partners from third sector organisations, including Down syndrome Scotland, and people with intellectual disabilities, who approved the work plan for this project prior to it commencing.

**Table 2** Characteristics of the cohorts

| Variable | Category | Data set 1 and 2 intellectual disabilities N=22 538 (100%) | Data set 1 and 2 autism N=27 741 (100%) | Data set 1 15% of general population* N=700 437 (100%) | Data set 2 5% of general population* N=233 378 (100%) |
|---|---|---|---|---|---|
| Sex | Male | 12 837 (57.0) | 21 390 (77.1) | 335 226 (47.9) | 111 497 (47.8) |
| | Female | 9701 (43.0) | 6351 (22.9) | 365 211 (52.1) | 121 881 (52.2) |
| Age | 0–15 | 4509 (20.0) | 15 387 (55.5) | 117 980 (16.8) | 39 220 (16.8) |
| | 15–24 | 3546 (15.7) | 6790 (24.5) | 76 184 (10.9) | 25 574 (11.0) |
| | 25–34 | 2976 (13.2) | 2024 (7.3) | 83 868 (12.0) | 27 721 (11.8) |
| | 35–45 | 3277 (14.5) | 1300 (4.7) | 97 886 (14.0) | 32 599 (14.0) |
| | 45–54 | 3664 (16.3) | 1053 (3.8) | 108 052 (15.4) | 36 164 (15.5) |
| | 55–64 | 2494 (11.1) | 617 (2.2) | 92 698 (13.2) | 31 022 (13.3) |
| | 65–74 | 1330 (5.9) | 300 (1.1) | 67 532 (9.6) | 22 166 (9.5) |
| | 75+ | 742 (3.3) | 270 (1.0) | 56 237 (8.0) | 18 912 (8.1) |
| Ethnicity† | White | 22 022 (97.7) | 26 976 (97.2) | Not available | 225 626 (96.7) |
| | Asian/Asian Scottish/Asian British | 368 (1.6) | 425 (1.5) | Not available | 5181 (2.2) |
| | African | 35 (0.2) | 80 (0.3) | Not available | 928 (0.4) |
| | Caribbean or black | 21 (0.1) | 24 (0.1) | Not available | 213 (0.1) |
| | Mixed/multiple ethnic groups | 66 (0.3) | 189 (0.7) | Not available | 888 (0.4) |
| | Other ethnic group | 26 (0.1) | 47 (0.2) | Not available | 542 (0.2) |
| SIMD | 1—most deprived | 6200 (27.5) | 6748 (24.3) | 130 354 (18.6) | 42 828 (18.4) |
| | 2 | 5481 (24.3) | 5901 (21.3) | 135 311 (19.3) | 45 407 (19.5) |
| | 3 | 4516 (20.0) | 5678 (20.5) | 142 238 (20.3) | 47 886 (20.5) |
| | 4 | 3693 (16.4) | 5128 (18.5) | 147 866 (21.1) | 49 429 (21.2) |
| | 5—most affluent | 2648 (11.7) | 4286 (15.5) | 144 668 (20.7) | 47 828 (20.5) |
| General health status† | Very good | 3389 (15.0) | 10 510 (38.0) | Not available | 122 511 (52.4) |
| | Good | 7942 (35.2) | 9608 (34.6) | Not available | 69 575 (30.0) |
| | Fair | 7715 (34.2) | 5387 (19.4) | Not available | 28 273 (12.1 |
| | Bad | 2356 (10.5) | 1533 (5.5) | Not available | 9885 (4.2) |
| | Very bad | 1136 (5.0) | 703 (2.5) | Not available | 3134 (1.3) |
| Mental health condition† | Yes | 4755 (21.1) | 3998 (14.4) | Not available | 9710 (4.2) |
| | No | 17 783 (78.9) | 23 743 (85.6) | Not available | 223 668 (95.8) |

*General population with neither intellectual disabilities nor autism.
†Data not available for data set 1 to ensure data minimisation, as it does not have sufficient power to investigate these factors in relation to causes of deaths and cancers.
SIMD, Scottish Index of Multiple Deprivation.

## FINDINGS TO DATE

To date, research has been completed on the cross-sectional Scottish Census, 2011 data only. We have analysed, compared with the general population, the independent contributions of having intellectual disabilities, and being autistic, on general health. We have also reported general health comparisons between the general population, with autistic people who also have intellectual disabilities. Further, we have reported mental health data compared with the general population on the people with intellectual disabilities (whether or not they were also autistic); autistic people (whether or not they also had intellectual disabilities); and autistic people who also have intellectual disabilities. We then reported on visual and hearing impairments, and physical disabilities in autistic people (whether or not they also had intellectual disabilities); and in autistic people who also had intellectual disabilities. These distinctions are important given the high proportion of overlap between intellectual disabilities and autism.

We derived a dichotomised variable of poor general health from the five response general health question in

Scotland's Census, 2011. In comparison with the general population, in children/youth, both intellectual disabilities (OR=18.34 (95% CI 917.17 to 19.58)), and autism (OR=8.40 (95% CI 8.02 to 8.80)) were independently associated with poor general health. In adults, both intellectual disabilities (OR=7.54 (95% CI 7.02 to 8.10)), and autism (OR=4.46 (95% CI 4.06 to 4.89)) were also independently associated with poor general health.[41] People with co-occurring intellectual disabilities and autism combined were particularly vulnerable to poor general health.[42]

In comparison with the general population, mental health conditions (answered as 'yes' or 'no') were also more common in people with intellectual disabilities (OR=7.1 (95% CI 6.8 to 7.3)),[43] autistic adults (OR=8.6 (95% CI 8.2 to 9.0))[44] and people with co-occurring intellectual disabilities and autism (OR=130.8 (95% CI 117.1 to 146.1)).[45] Poor general health was also associated with poor mental health in people with intellectual disabilities.[43]

Visual impairment, hearing impairment and physical disability were also more common than in the general population. For autistic adults, and autistic children/youth, respectively, visual impairment (OR=8.5 (95% CI 7.9 to 9.2) and OR=8.9 (95% CI 8.1 to 9.7)); hearing impairment (OR=3.3 (95% CI 3.1 to 3.6) and OR=5.4 (95% CI 5.1 to 5.6)); and physical disability (OR=6.2 (95% CI 5.8 to 6.6) and OR=15.8 (95% CI 14.1 to 17.8)) were considerably more common.[44 46] For people with both intellectual disabilities and autism, visual impairment (OR=65.9 (95% CI 58.7 to 73.9)); hearing impairment (OR=22.0 (95% CI 19.2 to 25.2)); and physical disability (OR=157.5 (95% CI 144.6 to 171.7)) were even more common.[44]

Raising awareness of these issues is important, as these conditions are disabling, and in some cases life limiting if left untreated. In addition to the communication needs that people with intellectual disabilities and autistic people have due to their neurodevelopmental condition, comorbidity renders differential diagnosis and treatments more complex, particularly if unrecognised, hence it is essential that clinicians are aware of the additional conditions that their patients may have.

### STRENGTHS AND LIMITATIONS

Strengths of the e-cohorts include their large scale, national Census coverage of the whole country of Scotland, with 94% uptake, and a high level of successful record linkage (>92%), hence limiting bias, and the systematic enquiry about intellectual disabilities and autism of the whole population. The questions on intellectual disabilities and autism were subject to cognitive question testing prior to use, to ensure that they accurately captured the conditions, and were acceptable to the population. Additionally, specific learning disabilities were distinguished from intellectual disabilities or autism, which is important given lay terms in current use,

for example, in the UK and Ireland. We are not aware of any other national Census data which has included data on these conditions, other than Irish data which has the limitation of not distinguishing intellectual disabilities from specific learning disabilities. Scotland has a rich profile of routinely collected health data which has been successfully used many times in previous research on the general population, and linkage of Scotland's Census, 2011 to these data, has provided large scale, whole country cohorts of intellectual disabilities, and of autism.

Limitations include that routinely collected data lack disease-specificity and granularity, and cause of death data are infrequently verified by post-mortem. Ascertainment of intellectual disabilities and autism was based on self or proxy reporting. Scotland is a high-income country with good country-wide educational and diagnostic services, and diagnosis of neurodevelopmental conditions attracts additional support for learning so is advantageous for the child. If a child's learning is delayed, schools have a statutory responsibility to seek assessment, so such children will have received a formal diagnosis. The families of children with learning delay are integral to the assessment and diagnostic care pathway for the child, and the development of the co-ordinated support plan for their additional support needs. Proxy reporting is the basis for much of the healthcare provided for people with intellectual disabilities and autistic people who cannot self-report, and overall, proxy reports are concluded to be a useful addition to determine aspects of well-being when the need arises.[47] Those adults for whom a diagnosis of autism was recorded will reflect the contemporary diagnostic practices in place during their childhood; and the concept of the autism spectrum has broadened in recent years.

### COLLABORATION

The breadth of the data sets created means they are suitable to be repurposed to answer many additional questions to the ones we outline in our current programme of research. Collaboration is encouraged, and as we are not the data controllers, this would be subject to approval by Scotland's Public Benefit and Privacy Panel for Health, Scotland's Statistics Public Benefit and Privacy Panel and ethical approval by an appropriate ethical committee.

**Acknowledgements** The authors would like to acknowledge the support of the eDRIS Team (Public Health Scotland), in particular David Clark, for their involvement in obtaining approvals, provisioning and linking data and for the use of the secure analytical platform within the National Safe Haven.

**Contributors** S-AC conceived and designed the programme, and wrote the first draft of the manuscript. AH and DK contributed to the design, and project management. DM, MF and GSS designed and supervised the statistics. LAH-M, ER and KD analysed data. JPP contributed expertise on data linkage and data sets. CM oversaw programme management. All contributed to drafting the manuscript, and approved the final version. S-AC is the guarantor.

**Funding** This work was supported by the UK Medical Research Council, grant number: MC_PC_17217, the Scottish Government via the Scottish Learning Disabilities Observatory and the Baily Thomas Charitable Fund.

**Competing interests** None declared.

**Patient and public involvement** Patients and/or the public were involved in the design, or conduct, or reporting, or dissemination plans of this research. Refer to the Cohort Description section for further details.

**Patient consent for publication** Not applicable.

**Ethics approval** This programme of research was approved by Scotland's Public Benefit and Privacy Panel for Health (reference: 1819–0051), Scotland's Statistics Public Benefit and Privacy Panel (1819–0051), and the University of Glasgow's College of Medical, Veterinary, and Life Sciences Ethical Committee (reference: 200180081).

**Provenance and peer review** Not commissioned; externally peer reviewed.

**Data availability statement** Data may be obtained from a third party and are not publicly available. Data access is available through approval by Scotland's Public Benefit and Privacy Panel for Health, Scotland's Statistics Public Benefit and Privacy Panel, and an appropriate ethical committee.

**ORCID iDs**
Sally-Ann Cooper http://orcid.org/0000-0001-6054-7700
Angela Henderson http://orcid.org/0000-0002-6146-3477
Gillian S Smith http://orcid.org/0000-0001-7241-6951
Laura Anne Hughes-McCormack http://orcid.org/0000-0001-9498-8045

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
