## [Reviewer comments · BMJ Open]

ARTICLE DETAILS

TITLE (PROVISIONAL)	Cohort profile: Scotland's record-linkage e-cohorts of people with intellectual disabilities, and autistic people (SCIDA)
AUTHORS	Cooper, Sally-Ann; Henderson, Angela; Kinnear, Deborah; Mackay, Daniel; Fleming, Michael; Smith, Gillian; Hughes-McCormack, Laura; Ryzewska, Ewelina; Dunn, Kirsty; Pell, J. P.; Melville, Craig

VERSION 1 – REVIEW

REVIEWER	Bourke, Jenny Telethon Kids Institute, Epidemiology
REVIEW RETURNED	29-Nov-2021

GENERAL COMMENTS	The paper provides a clear description of the cohort of people with intellectual disability or autism, and the comparison groups, defined through the Scotland Census in 2011. The introduction gives a comprehensive background on the health disparities for people with intellectual disability or autistic people. The sentence page 5, line 9 describing the population numbers is a little ambiguous in whether the 5,295,403 describes the total population number or the respondents, and could be clarified. The number of individuals with intellectual disability overall (0.5%) seems a little low compared with population prevalence (eg Maulik 2011). However this may be explained by the self/proxy-report of intellectual disability. The number with autism is greater (0.6%) but as explained in the paper the autistic individuals are predominantly in the lower age groups. It would have been interesting to know the proportion of the autistic people who identified as having co-morbid intellectual disability and I presume these groups are mutually exclusive? Some clarification on whether individuals with intellectual disability and autism are only in the autism group or not would be worth clarifying. It is not clear why the data on ethnicity and health status for the 15% comparison group were not available. The power calculation figures provide a clear insight into how the size of the comparison groups were determined. Overall the paper will provide a convenient reference for the future analyses to be undertaken which will provide important information from the research questions posed.
--

REVIEWER	Lunsky, Yona Centre for Addiction and Mental Health, Underserved Populations Program
REVIEW RETURNED	29-Nov-2021

GENERAL COMMENTS

This paper describing the unique ID and autism cohorts of Scottish children and adults is a very important contribution to the field of intellectual disability/autism and health. These cohorts are the only known population cohorts which combine administrative health data with census data for an entire country.

I think it is helpful to have an overview paper which brings together the methodology and description of cohorts for the valuable e-cohorts assembled through the linkage of Scotland's census from 2011 and routinely collected health, death and health care data.

This paper provides estimated prevalence of ID and of autism, it also describes for some demographics, how people with both autism and ID present, relative to the general population.

1. I wonder if the paper might be clearer if the purpose in both the abstract and the paper itself could differentiate between the purpose of the overall linkage project and the purpose of this paper, which as I see it, is an outline of the effort, the demographic descriptions of two cohorts, and a description of variables which are available to study in the future because of this linkage.

2. In the literature review or in the strengths section of the paper within the discussion, it would be important to demonstrate how many other studies have been able to identify almost the entire population through census data, and why looking for diagnostic information through existing health records is a more limiting approach. The current review focuses specifically on the health and health care of the population without discussing how information has been gleaned in prior studies.

3. On page 5 of 24, when the authors present census rates of autism and ID, or in tables, it might be important to offer sex stratified information.

4. At the bottom of page 6 of 24 in the cohort description, they use the term developmental disorder and offer examples which are under the autism spectrum. It is not clear whether this term was selected based on feedback from autistic adults and family caregivers, or whether another term was used. Throughout, the authors use the term autism, but was the term in the census was developmental disorder (which could have been broader)? I think noting that the terms used throughout the paper are intellectual disability and autism would be appropriate here.

5. From the description of available datasets, it is not clear how diagnostic information available in the Tables has been captured. Was general health status and mental health condition (Y/N) in the census? Has there been any comparison of diagnostic information to see how consistent diagnostic information is from health usage data versus self report?

6. The result section (Findings to date) summarizes ratings of overall health and presence of mental health conditions in different subgroupings, as well as additional disabilities. This information is divided into odds ratios for children and adults and is sometimes presented separately for those with ID and for autistic people. However the third paragraph does not report on ID only; it reports on autism with and without ID.

Perhaps an overview paragraph at the start of findings thus far could orient the reader to the types of studies completed thus far, noting which cohorts were used for the analyses and how they were stratified. Given the emphasis on autism with and without ID in some of the papers, it may be important to highlight that the census information included some people who endorsed both “learning disability” and “developmental disorder”(autism; and present the percentage). But it is possible that some people in the learning disability/ID cohort may have also been autistic and vice versa.

7. It would also be important to outline in the results section the types of analyses underway to address the other objectives outlined on page 6.

8. The final section of the paper focuses on strengths and limitations and is fairly brief. It would be helpful here to differentiate between strengths and limitations of these datasets in terms of the analyses completed and findings already published, as well as strengths and limitations of what is to be studied next (e.g., limitations from cause of death data). Referring to literature outside of this cohort would be important here. There are some valuable papers on health data linkage (e.g., a special issue on the topic in IDD – see <https://pubmed.ncbi.nlm.nih.gov/31568741/> for an overview specific to US and this paper summarizes lessons from the Canadian and Australian experiences <https://pubmed.ncbi.nlm.nih.gov/31568733/>)

9. One strength identified was the separation of “Specific learning disabilities” from autism and intellectual disability. I wonder if this is only a concern in the UK where the term learning disability is used instead of intellectual disability, and may need further explanation. This is not typically listed as a limitation in studies from other countries because the diagnoses are not using the same terminology.

10. The sentences following the statement at the bottom fo page 8 of 24 about supports for learning is not clear. Is it suggesting that the authors are confident that self diagnosis is accurate because the country invests in early interventions and diagnoses of children? This would require further explanation.

11. One very important limitation which requires further explanation (and references) is that the diagnosis of autism is based on who was thought to be autistic in 2011 in Scotland. This would miss many females, and also would be quite heavily weighted toward the youngest people. This is quite evident from the Table 2. Importantly, when statements are made about the health of autistic people, because most of the people included are young, then the proportion of people in poor health is biased toward young people who have lower rates of chronic diseases which develop as a person ages.

Minor point in the literature review

12. Third line from bottom of first paragraph: these inequalities are or this inequality is

13. last paragraph of page 4 of 24 the article cited (#16) shows that adults with ID are reported to participate less in both cervical and breast cancer screening

VERSION 1 – AUTHOR RESPONSE

Reviewer: 1

Ms. Jenny Bourke, Telethon Kids Institute

Comments to the Author:

The paper provides a clear description of the cohort of people with intellectual disability or autism, and the comparison groups, defined through the Scotland Census in 2011. The introduction gives a comprehensive background on the health disparities for people with intellectual disability or autistic people.

The sentence page 5, line 9 describing the population numbers is a little ambiguous in whether the 5,295,403 describes the total population number or the respondents, and could be clarified.

Response: We have clarified this as follows : “It included data on 94% of the whole Scottish population, and imputed the remaining 6% (5,295,403 people in total).”**

The number of individuals with intellectual disability overall (0.5%) seems a little low compared with population prevalence (eg Maulik 2011). However this may be explained by the self/proxy-report of intellectual disability.

*****Response: We respectfully disagree; we do not think Scotland’s Census, 2011 under-reports the rate of intellectual disabilities. The over-arching figure quoted by Maulik et al (2011) was 10.37/1,000 from their review of 52 prevalence studies. However, most of the included studies were of children, and rates were higher for populations which only included children/adolescent populations, at 18.3/1,000, and lowest in adult only populations at 4.94/1,000 (and influenced by an outlier study from Iran with a much higher rate than the other adult studies). There are likely to be many reasons for the lower rates reported in adults, including the 20 year earlier age of death that they experience than in the general population, and their acquisition of skills and learning over time such that an intellectual impairment (IQ<70) which delays academic development in children, may no longer meet standard definitions of intellectual disabilities in adulthood (IQ<70, plus requiring additional support for daily activities). Most of the studies included in the review were of children +/- adolescent populations, as such studies on adults are highly complex and resource intensive to conduct, so few exist. As typically only about 15-20% of the population are children/adolescents, the over-arching prevalence figure is skewed as most studies are at that age range. In our own longitudinal work with data from Scotland’s annual pupil census, we have also found that some children who are classified as having additional needs due to intellectual disabilities in one year are “undiagnosed” in subsequent years i.e children move in and out of this categories, likely to be a reflection of a range of social factors that can impact on their educational development, and suggesting that rates of intellectual disabilities in children are over-reported. Our work with Scottish primary care data has also found a similar adult rate for intellectual disabilities as that we report in this paper from Scotland’s Census.

Maulik et al’s study must also be interpreted in the light that 25 of the studies did not provide the exact age range studied, a further two did not report their observation period, and prevalence varied not just according to age, but also by income group of the country of origin, with lower rates in high income countries. *****

The number with autism is greater (0.6%) but as explained in the paper the autistic individuals are predominantly in the lower age groups. It would have been interesting to know the proportion of the autistic people who identified as having co-morbid intellectual disability and I presume these groups are mutually exclusive? Some clarification on whether individuals with intellectual disability and autism

are only in the autism group or not would be worth clarifying.

*****Response: We have clarified this by adding in the introduction that: “This also accounts for why, of the autistic children and young people, 15.0% additionally had intellectual disabilities, whilst of the autistic adults aged 25 years and over, 29.4% additionally had intellectual disabilities. In total, 1.08/1,000 people had both autism and intellectual disabilities, and are represented in both the intellectual disabilities and the autism cohorts.”*****

It is not clear why the data on ethnicity and health status for the 15% comparison group were not available.

*****Response: We have clarified this by adding the following to the introduction: “The minimal number of variables was requested for each dataset: in dataset 1, this therefore did not include ethnicity, nor general or mental health variables, as there is not sufficient power to investigate these factors in relation to causes of deaths and cancers.” For clarity, we have also expanded the information in the legend below table 2 as follows: “Data not available for Dataset 1 to ensure data minimisation, as it does not have sufficient power to investigate these factors in relation to causes of deaths and cancers.”*****

The power calculation figures provide a clear insight into how the size of the comparison groups were determined.

Overall the paper will provide a convenient reference for the future analyses to be undertaken which will provide important information from the research questions posed.

*****Thank you for reviewing the paper, enabling us to improve it.*****

Reviewer: 2

Dr. Yona Lunsy, Centre for Addiction and Mental Health, University of Toronto

Comments to the Author:

This paper describing the unique ID and autism cohorts of Scottish children and adults is a very important contribution to the field of intellectual disability/autism and health. These cohorts are the only known population cohorts which combine administrative health data with census data for an entire country.

I think it is helpful to have an overview paper which brings together the methodology and description of cohorts for the valuable e-cohorts assembled through the linkage of Scotland’s census from 2011 and routinely collected health, death and health care data.

This paper provides estimated prevalence of ID and of autism, it also describes for some demographics, how people with both autism and ID present, relative to the general population.

1. I wonder if the paper might be clearer if the purpose in both the abstract and the paper itself could differentiate between the purpose of the overall linkage project and the purpose of this paper, which as I see it, is an outline of the effort, the demographic descriptions of two cohorts, and a description of variables which are available to study in the future because of this linkage.

*****Response: We agree and have added the following to the abstract: “This paper describes the cohorts, record-linkages, and variables that will be used”, and made other minor edits to retain the abstract within the required word-count. We have also clarified at the end of the introduction as follows: “The purpose of this report is to describe the e-cohorts we have created, the record-linkages that have been made, and the variables that are available within the datasets to investigate the above research questions.”*****

2. In the literature review or in the strengths section of the paper within the discussion, it would be important to demonstrate how many other studies have been able to identify almost the entire population through census data, and why looking for diagnostic information through existing health records is a more limiting approach. The current review focuses specifically on the health and health care of the population without discussing how information has been gleaned in prior studies.

****Response: We agree, and have added the following paragraph to the introduction:

“Internationally, one of the challenges in investigating these issues at a population level is the lack of data on representative samples of people with intellectual disabilities, or autistic people. Important studies have used novel methods to link routinely collected administrative data, and identify people with intellectual disabilities via their use of educational services, services designed for adults with intellectual disabilities and/or through identifying diagnostic codes in health records likely to indicate the presence of intellectual disabilities or autism.³²⁻³⁵ These approaches provide valuable data and also challenges, as, after leaving education/school, not all adults with intellectual disabilities use intellectual disabilities services, and nor do autistic adults, and there are coding issues in health records which are completed by multiple professionals and may be both incomplete and overinclusive for syndromes that are associated with intellectual disabilities in some but not all people. Additionally, caseness may be variably defined, with the term intellectual and developmental disorders being commonly used and which may or may not be a broader definition of neurodevelopmental disorders combining for example some or all of intellectual disabilities, autism, attention deficit hyperactivity disorder, and specific learning disabilities.³⁵ A further approach to population studies is using national surveys conducted with samples of the population, but has the limitation that people with intellectual disabilities or autism occur in only small numbers and so restricts the analyses that can be undertaken, and in longitudinal research they tend to have higher cohort attrition and more biased cohort retention than other people.^{35,36} In Scotland, the national Census in 2011 included questions on both intellectual disabilities and on autism, and so presents an ideal data-source for research with these populations. We are not aware of any other national Census data which has included data on intellectual disabilities and autism, other than Irish data which has the limitation of not distinguishing intellectual disabilities from specific learning disabilities.”

We have also added the following to the strengths and limitations section: “We are not aware of any other national Census data which has included data on these conditions, other than Irish data which has the limitation of not distinguishing intellectual disabilities from specific learning disabilities.”****

3. On page 5 of 24, when the authors present census rates of autism and ID, or in tables, it might be important to offer sex stratified information.

****Response: We agree and have added to this paragraph as follows: “In Scotland’s Census, 2011, of the whole Scottish population, 0.5% were recorded as having intellectual disabilities (0.4% females and 0.6% males), and 0.6% were recorded as being autistic people (0.3% female and 1.0% male). Regarding autism, 1.6% of all Scottish children and young people (<25 years of age) were recorded to have autism (0.7% females and 2.5% males), whilst 0.2% of adults aged 25 years and over were recorded to have autism (0.1% females and 0.3% males), reflecting the diagnostic broadening of autism spectrum disorders in recent years, and greater awareness of the condition.”****

4. At the bottom of page 6 of 24 in the cohort description, they use the term developmental disorder and offer examples which are under the autism spectrum. It is not clear whether this term was selected based on feedback from autistic adults and family caregivers, or whether another term was used. Throughout, the authors use the term autism, but was the term in the census was developmental disorder (which could have been broader)? I think noting that the terms used throughout the paper are intellectual disability and autism would be appropriate here.”

****Response: We have added more information to the process leading to the wording the Census questions as follows: “.....This resulted in a redesign of the question on autism, to the version included in the actual Census, to accurately capture the data on autism, in keeping with feedback from the pilot testing with people with autism. (Henceforth, throughout this paper we refer to

“autism”, rather than the exact wording on the census described above). Importantly, question 20 distinguished autism from learning disability (which in the UK is synonymous to the international term ‘intellectual disabilities’),.....”

5. From the description of available datasets, it is not clear how diagnostic information available in the Tables has been captured. Was general health status and mental health condition (Y/N) in the census? Has there been any comparison of diagnostic information to see how consistent diagnostic information is from health usage data versus self report?

****Response: We have clarified as follows: “We derived a dichotomised variable of poor general health from the five response general health question in Scotland’s census, 2011.”, and “In comparison with the general population, mental health conditions (answered as “yes” or “no”).....” We have not undertaken further analyses comparing Census data with the newly linked other datasets, as work on the linked dataset has only just began. The first sentence highlights that we have, at present only analysed the cross-sectional Census data: “To date, research has been completed on the cross-sectional Scottish Census, 2011 data only.”

6. The result section (Findings to date) summarizes ratings of overall health and presence of mental health conditions in different subgroupings, as well as additional disabilities. This information is divided into odds ratios for children and adults and is sometimes presented separately for those with ID and for autistic people. However the third paragraph does not report on ID only; it reports on autism with and without ID.

Perhaps an overview paragraph at the start of findings thus far could orient the reader to the types of studies completed thus far, noting which cohorts were used for the analyses and how they were stratified. Given the emphasis on autism with and without ID in some of the papers, it may be important to highlight that the census information included some people who endorsed both “learning disability” and “developmental disorder”(autism; and present the percentage). But it is possible that some people in the learning disability/ID cohort may have also been autistic and vice versa.

****Response: We have added clarification at the start of this section as follows: “We have analysed, compared with the general population, the independent contributions of having intellectual disabilities, and being autistic, on general health. We have also reported general health comparisons between the general population, with autistic people who also have intellectual disabilities. Further, we have reported mental health data compared with the general population on the people with intellectual disabilities (whether or not they were also autistic); autistic people (whether or not they also had intellectual disabilities); and autistic people who also have intellectual disabilities. We then reported on visual and hearing impairments, and physical disabilities in autistic people (whether or not they also had intellectual disabilities); and in autistic people who also had intellectual disabilities. These distinctions are important given the high proportion of overlap between intellectual disabilities and autism.”*****

7. It would also be important to outline in the results section the types of analyses underway to address the other objectives outlined on page 6.

****Response: We have not done this, as the journal guidelines to authors for cohort papers advise for the “findings to date” section: “Include a short explanation of the most notable results from the cohort so far, with references to relevant publications. This section should summarise rather than present results.”*****

8. The final section of the paper focuses on strengths and limitations and is fairly brief. It would be helpful here to differentiate between strengths and limitations of these datasets in terms of the analyses completed and findings already published, as well as strengths and limitations of what is to be studied next (e.g., limitations from cause of death data). Referring to literature outside of this cohort would be important here. There are some valuable papers on health data linkage (e.g., a special issue on the topic in IDD – see <https://pubmed.ncbi.nlm.nih.gov/31568741/> for an overview

specific to US and this paper summarizes lessons from the Canadian and Australian experiences <https://pubmed.ncbi.nlm.nih.gov/31568733/>)

*****Response: We have written the strengths and limitations section with regards to the strengths and limitations of the investigations we are now undertaking, as the previous work was only based on Scotland Census, 2011 data, not the record-linked data with these e-cohorts we describe, that forms the basis of the future studies. (The published papers we quote from the cross-sectional Scotland Census, 2011 analyses all include discussion on their strengths and limitations.) We acknowledge the limitation of cause of death data. We agree these are important references to include, and have done so in the new paragraph added to the introduction.

9. One strength identified was the separation of “Specific learning disabilities” from autism and intellectual disability. I wonder if this is only a concern in the UK where the term learning disability is used instead of intellectual disability, and may need further explanation. This is not typically listed as a limitation in studies from other countries because the diagnoses are not using the same terminology.

*****Response: We have retained this as we strongly believe it to be a key strength of the cohort identification and internationally language in this area is continually evolving and clarity is important. In recognition of the reviewer’s comment we have added to the relevant sentence as follows: “Additionally, specific learning disabilities were distinguished from intellectual disabilities or autism, which is important given lay terms in current use e.g. in the UK and Republic of Ireland.”*****

10. The sentences following the statement at the bottom of page 8 of 24 about supports for learning is not clear. Is it suggesting that the authors are confident that self diagnosis is accurate because the country invests in early interventions and diagnoses of children? This would require further explanation.

*****Response: We have clarified that this is not self diagnosis, by adding the following information: “The families of children with learning delay are integral to the assessment and diagnostic care pathway for the child, and the development of the co-ordinated support plan for their additional support needs.”*****

11. One very important limitation which requires further explanation (and references) is that the diagnosis of autism is based on who was thought to be autistic in 2011 in Scotland. This would miss many females, and also would be quite heavily weighted toward the youngest people. This is quite evident from the Table 2. Importantly, when statements are made about the health of autistic people, because most of the people included are young, then the proportion of people in poor health is biased toward young people who have lower rates of chronic diseases which develop as a person ages.

*****Response: We agree that the diagnostic criteria for autism have broadened considerably in recent years (we do refer to this in the introduction: “Regarding autism, 1.6% of all Scottish children and young people (<25 years of age) were recorded to have autism (0.7% females and 2.5% males), whilst 0.2% of adults aged 25 years and over were recorded to have autism (0.1% females and 0.3% males), reflecting the diagnostic broadening of autism spectrum disorders in recent years, and greater awareness of the condition.”; and in the limitations section: “Those adults for whom a diagnosis of autism was recorded will reflect the contemporary diagnostic practices in place during their childhood; and the concept of the autism spectrum has broadened in recent years.”). This is why in our publications thus far, and our planned publications, we have and will always stratify findings by age. Autistic females may be more likely to be underdiagnosed than autistic males in people of average or mild intellectual disabilities, though this possibly might not be the case in people with more severe intellectual disabilities, so we have added to the introduction as follows: “There is some evidence to suggest that autism is less recognised in females than in males,³⁷ but less so for people with more severe intellectual disabilities.³⁸”*****

Minor point in the literature review

12. Third line from bottom of first paragraph: these inequalities are or this inequality is

****Response: This error has been corrected.****

13. last paragraph of page 4 of 24 the article cited (#16) shows that adults with ID are reported to participate less in both cervical and breast cancer screening

****Response: We have corrected this by adding: "and breast screening"****

****Thank you for reviewing the paper, enabling us to improve it.****

VERSION 2 – REVIEW

REVIEWER	Bourke, Jenny Telethon Kids Institute, Epidemiology
REVIEW RETURNED	07-Feb-2022

GENERAL COMMENTS	Thank you for responding to my queries with clarity. The paper should provide a useful description of this cohort for future analyses.
--

REVIEWER	Lunsky, Yona Centre for Addiction and Mental Health, Underserved Populations Program
REVIEW RETURNED	07-Feb-2022

GENERAL COMMENTS	This is a rich and thorough contribution to the literature. I appreciate the edits made in the resubmission to address reviewer concerns/questions.
---